# Challenges for Artificial Intelligence in Recognizing Mental Disorders

**DOI:** 10.3390/diagnostics13010002

**Published:** 2022-12-20

**Authors:** Wen-Jing Yan, Qian-Nan Ruan, Ke Jiang

**Affiliations:** 1Wenzhou Seventh People’s Hospital, Wenzhou 325005, China; 2School of Mental Health, Wenzhou Medical University, Wenzhou 325015, China; 3The Social Work Service Center of Zhuji, Zhuji 311800, China

**Keywords:** artificial intelligence, depressive disorder, mental disorder, dataset, diagnosis

## Abstract

Artificial Intelligence (AI) appears to be making important advances in the prediction and diagnosis of mental disorders. Researchers have used visual, acoustic, verbal, and physiological features to train models to predict or aid in the diagnosis, with some success. However, such systems are rarely applied in clinical practice, mainly because of the many challenges that currently exist. First, mental disorders such as depression are highly subjective, with complex symptoms, individual differences, and strong socio-cultural ties, meaning that their diagnosis requires comprehensive consideration. Second, there are many problems with the current samples, such as artificiality, poor ecological validity, small sample size, and mandatory category simplification. In addition, annotations may be too subjective to meet the requirements of professional clinicians. Moreover, multimodal information does not solve the current challenges, and within-group variations are greater than between-group characteristics, also posing significant challenges for recognition. In conclusion, current AI is still far from effectively recognizing mental disorders and cannot replace clinicians’ diagnoses in the near future. The real challenge for AI-based mental disorder diagnosis is not a technical one, nor is it wholly about data, but rather our overall understanding of mental disorders in general.

## 1. Introduction

Today, artificial intelligence has a wide range of applications in medical diagnosis (e.g., the use of computer vision to classify images that can identify tumors [1], ulcers [2], etc.), often with higher accuracy than professional doctors. In recent years, more and more research has focused on the diagnosis of psychiatric disorders, seemingly achieving significant results. A mental disorder is a behavioral or mental pattern that causes significant distress or impairment of personal functioning [3], such as major depressive disorder (MDD). Hundreds of articles are published every year about the use of machine learning to predict depression. The average accuracy rate found in recent years has been approximately 80%, and there are many studies reporting accuracy rates above 90% [4]. For example, researchers using the transformation of EEG features and machine learning methods obtained an 89.02% accurate classification and concluded that, in the future, it will be possible for EEG-based portable system design and applications to be developed for auxiliary depression recognition [5]. As many sensors are becoming more portable and even wearable, depression monitoring and prediction through AI has increased in popularity and could be conducted at any time in daily life.

In contrast, psychiatrists are often overworked, and their consistency and accuracy are low (i.e., their misdiagnosis rate is high) because diagnoses mainly depend on personal experience and subjective evaluation, with no significant bio-markers. One study conducted in a specialized psychiatric setting in Ethiopia revealed that more than one-third of patients with severe psychiatric disorders were misdiagnosed (39.16%) [6]. The misdiagnosis rate is even higher for non-psychiatrists. Of the 840 primary care patients assessed in seven primary care clinics in three Canadian provinces, misdiagnosis rates reached 65.9% for major depressive disorder, 92.7% for bipolar disorder, 85.8% for panic disorder, 71.0% for generalized anxiety disorder, and 97.8% for social anxiety disorder [7]. The diagnosis of mental disorders almost exclusively depends on doctor–patient communication and scale analysis, which have obvious disadvantages such as patient denial, poor sensitivity, subjective biases, and inaccuracy. Moreover, there is a critical shortage of psychiatrists in many regions, especially in developing countries, making it impossible to meet local needs. In contrast, using machines to diagnose mental illnesses offers many benefits, such as saving human resources, increasing efficiency, achieving large-scale assessments, and reducing the stigma of illness. Therefore, an objective, automated method for helping psychiatrist to diagnose mental disorders is becoming increasingly necessary.

However, today we see few large-scale applications of AI to detect mental illness. Why? First of all, it is important to understand the ways in which AI automates diagnosis.

## 2. The Rationale for Using Machine Learning to Identify Mental Disorders

Since the term “mental disorder” includes different categories with high heterogeneity, we mainly use the most common category of mental disorders, depressive disorders, as an example in the present research. Previous studies have found differences between depressed patients and the healthy population. These can include biochemical indicators such as blood oxygen consumption in the brain [8], neurotransmitters [9], and EEG [10], peripheral physiological signals [11] such as heart rate, skin conductance, etc., and non-verbal behaviors [12] such as facial expressions and voice features, the language used (verbal or textual), etc. The features that differ between the two groups (those with and without a depressive disorder) serve as basic AI classifiers. Therefore, by using these discriminant metrics as feature inputs for machine learning, it should be possible to train good predictive models for automated depressive disorder diagnosis.

Many studies have used nonverbal behaviors to predict depression, especially facial expressions [13,14], which are the most salient of behaviors and are considered to accurately display mood (as depressive disorders are mood disorders). We chose facial cues as an example for the present research, but the application of other cues can be found in certain survey articles [5,8,15]. Facial expressions, usually categorized as expressing anger, sadness, joy, surprise, disgust, fear, etc., are regarded as discriminative cues for depressive disorder detection. Those diagnosed with a depressive disorder often demonstrate little expressiveness in their facial expressions [16]. Gavrilescu et al. proposed the determination of depression levels by analyzing facial expressions via the Facial Action Coding System. The experiment obtained 87.2% accuracy for depression identification [17]. Furthermore, the duration of spontaneous smiles [18], smile intensity [18,19], mouth animation [20], and lack of smile [21] have also been considered to offer valuable patterns for depressive disorder detection. In recent years, the use of facial expressions as cues for depression recognition has made great progress. Effective facial features can now even include pupil changes. For example, a recent study considered faster pupillary responses to represent a positive healthy control [22]. Depressed subjects demonstrate slower pupil dilation responses in certain conditions [23]. One study found that pupil bias and diameter were important for assessing the symptoms of depression [24]. Features consist of reduced eye contact [21], gaze direction [19], eyelid activity, and eye movement and blinking [25].

Single modal (such as only facial cues) depressive disorder recognition has also been found to yield positive results. Theoretically, multimodal data should be able to further enhance the effect, such as when voice and visual cues are combined as feature input, and the addition of physiological information should further enhance the accuracy of automatic diagnosis. Multimodality is a prominent direction of both algorithm development and database development.

Researchers use visual, acoustic, verbal, and physiological signals to make predictions, so what exactly are the characteristics of these diseases they are attempting to predict? What is the diagnostic process followed by clinicians? Do these issues pose challenges to AI that are different from those seen with other tasks?

We use the example of depressive disorders to illustrate the challenges of using AI to diagnose mental illness. The DSM-5 [26] or ICD-11 [27] are the most authoritative diagnostic manuals available. However, even the DSM, the most widely used standard, is highly controversial. Next, we will discuss the features of depressive disorders that may pose a barrier to using AI diagnosis.

## 3. Challenges from Diagnostic Criteria

First, many diagnostic indicators are based on subjective experiences or qualitative descriptions or are difficult to objectively quantify and standardize. Diagnostic criteria for depressive disorders are based on symptomatology, such as a depressed mood or sleep problems. Although many scholars pursue a physiological basis or biomarkers, there are no clinically useful diagnostic biomarkers that are able to absolutely confirm a diagnosis of major depressive disorder.

Second, individuals vary greatly in their presentation of symptoms (see Figure 1). According to the DSM-5, the two most important core symptoms of major depressive disorder are (1) a depressed mood most of the day and/or (2) markedly diminished interest or pleasure. At least three or four more of the other seven need to be met to be diagnosed with a depressive disorder, meaning that depressive disorders themselves do not have consistent symptoms and vary greatly among individuals. The PHQ-9 (Patient Health Questionnaire-9) is an assessment of the nine criteria in the DSM-5. Another widely used scale, the HAMD, does not focus on typological symptoms (i.e., insomnia, low mood, agitation, anxiety, and reduced weight). In addition, differences in symptoms exist across developmental stages. For example, depressive symptoms in adolescents tend to manifest as irritability and not necessarily in a constant low mood. There exist, at fewest, 1497 unique profiles for depression [28]. In some cases, patients with the same diagnosis may not share any identical symptoms [29].

Third, depressive disorders comprise a collection of ailments with many subcategories and variants, such as disruptive mood dysregulation disorder, major depressive disorder, and persistent depressive disorder.

Fourth, co-morbidities are very common in mental disorders (see Figure 1). Depressive disorders may be accompanied by anxiety and personality disorders and are often confused with bipolar or other mental disorders. Such illnesses may be very similar or identical to particular depressive disorders in terms of symptoms (e.g., sleep or appetite problems) but require differential diagnoses by clinicians. This issue often results in subjective bias [30,31,32,33,34].

Fifth, the symptoms should lead to impaired social functioning according to DSM. They are expected to cause clinically significant distress or impairment in social, occupational, or other important areas of functioning that are culturally related.

Sixth, the symptoms are not static and not always displayed. Major depressive disorder is not continuous, but rather episodic. For example, some people feel more serious in the morning on one day and remain depressed for several weeks.

Seventh, depressive disorders manifest in interactions between genetic issues and environmental, physiological, and sociocultural factors. The pathogenesis of depressive disorders has not yet been unilaterally agreed upon. Depression is not just a neurophysiological problem. It depends on the interaction between a genetic predisposition and environmental factors [35]. The combination of biological elements, family and environmental stressors, and personal vulnerabilities plays a vital role in affecting the onset of major depressive disorder [36]. This makes the subjective experience of depression and the behavioral and speech characteristics of depressed individuals very different.

## 4. Challenges from Standard Diagnostic Approaches

Then, how do clinicians diagnose depressive disorders, considering the variability of such characteristics?

As described above regarding the qualities and diagnostic criteria of depressive disorders, there is no unique and efficient clinical set of indicators, making the diagnosis of depressive disorders time-consuming and inherently subjective [37]. Routine assessments include self-rating scales and clinician-based interviews. Both such assessments are mainly based on the DSM and ICD. Self-rating scales are a simple and convenient way to assess depressive disorders; examples include the PHQ-9, Zung’s Self-rating Depression Scale, and the Beck Depression Self-Rating Scale. The results are most often used for screening and providing a reference for physicians’ diagnoses. Self-rating scales have been used widely in various studies, with specificity and sensitivity reaching up to 80% to 90%, though there are certain problems [38]. In addition to self-rating assessments, other rated scales such as the Hamilton Rating Scale for Depression [39] are often also used to assist clinicians’ diagnosis.

Clinical interviews are more professional and accurate but also more time-consuming and laborious. Doctors’ interview-based assessments comprise the final decision stage for diagnosis. Diagnosing depressive disorders can be complicated, depending not only on the educational background, cognitive ability, and honesty of the subject describing their symptoms but also on the experience and motivation of the clinician. Comprehensive information and thorough clinical training are needed to accurately diagnose the severity of depression [40]. Some biological markers such as low serotonin levels [41], neurotransmitter dysfunction [42], and brain structure [43] have been considered to be indicators of depression.

Depressive disorders are so complex that the diagnostic process must be considered holistically. Because depressive disorders are not just mood problems but also sociocultural in nature, they are often accompanied by a serious impairment of social functioning. This may explain why the misdiagnosis rate is high for clinicians. It requires us to rethink the current depression dataset and ask whether the samples are representative and qualified. Can the objective features recorded predict depressive disorders? Are the annotations valid?

## 5. Challenges from the Logical Fallacy of Mental Disorder Diagnosis

When a clinician diagnoses a person as having a depressive disorder, they rely on the symptoms reported by the client, such as a persistent low mood for two weeks and frequent suicidal thoughts. What is the cause of the persistent low mood? The usual answer is that the individual is suffering from a depressive disorder. Depressive disorders cause the corresponding symptoms, which is the premise of conditional reasoning: If p then q. Diagnosing disorders by symptoms requires the reasoning of affirming the consequent: If q then p. This is a logical fallacy (Table 1).

Depressive disorders are labels for sets of symptoms. This means that symptoms do not explain why a person has a depressive disorder, nor does the disorder explain why the symptoms occur. Therefore, it is implausible to identify whether a person has a depressive disorder from the symptoms they present. A person may be depressed because they have been experiencing negative stimuli for the past several weeks. If the negative stimuli disappear, so might the negative mood. On the other hand, they may be depressed because of another psychological disorder, such as a personality disorder.

We are not yet sure of the relationship between depressive disorders and the set of symptoms, which causes significant problems with diagnosis. If AI tries to imitate clinicians, it will face the same challenge.

## 6. Challenges from Datasets

Computer scientists put great effort into training computationally fast and robust models based on specific databases. Therefore, the accuracy of depressive disorder recognition is largely limited by the quality of the samples and their annotation in the datasets. Below, we introduce basic information about the datasets and examine problems stemming from the characteristics and diagnostic criteria of depressive disorders.

### 6.1. Example Datasets for Depression Recognition

Most of the existing databases have only been used for the research for which they were created and have not been released publicly for depression recognition studies. Only a few databases have been released publicly for depression recognition purposes. Some famous ones include AVEC (The Continuous Audio/Visual Emotion and Depression Recognition Challenge) 2013, AVEC2014, and DAIC-WOZ (Distress Analysis Interview Corpus-Wizard of Oz).

The AVEC2013 and AVEC2014 datasets are a subset of the Audio-Visual Depression Language Corpus. AVEC2013 [44] is comprised of 340 videos recorded in German, in which participants performed human–computer interaction tasks in front of a webcam and microphone. The video files include free speech, reading, singing, and picture-based association tasks. BDI-II (Beck Depression Inventory II) [45] was used to annotate the depression severity score for each participant’s interview records. AVEC2014 [46] is a subset of AVEC2013, consisting of 300 videos recorded in German. The duration of each video clip is shorter than the clip appearing in AVEC2013. DAIC-WOZ [47] is part of the Distress Analysis Interview Corpus annotated by PHQ-8 [48]. It was employed for AVEC2016 and AVEC2017. DAIC-WOZ adopts a virtual interviewer, the emotional status of which is strictly controlled during the interview. Audio, video, and deep sensor modalities were collected for the dataset. It also contains information on galvanic skin response, electrocardiogram information, and participants’ respiratory data. The E-DAIC is an extended version of the DAIC-WOZ. The data were collected from semi-clinical interviews designed to support the diagnosis of psychological distress conditions such as anxiety and depression [15]. The dataset contains 163 development samples, 56 training samples, and 56 test samples; participants’ data include age and gender, and the PHQ-8 scores are labelled. This database was employed for AVEC2019 [21].

### 6.2. Are the Recorded Data Sufficiently Valid and Ecological?

Many samples are videos of individuals being interviewed face-to-face by clinical doctors or virtual interviewers. In some data collection processes, other modality information such as depression scale data, dynamic physiological information, etc., is also recorded at the same time for auxiliary analysis [15].

It is important to note that this is a special situation and the interviewee’s performance is quite important. A patient in an outpatient clinic must demonstrate their depression to make the clinician understand their status. They must describe their troubles prompted by the disease, appear sad, and perhaps allow despair to come through in their voice, even though this might not accurately reflect their current state. Major depressive disorder is not continuous, but rather episodic. A patient may need to “act” as if they are in a depressed state, different from healthy people and their daily life. Therefore, the samples of patients’ performances are deliberate and supposed to be different from their normal status. This also means that the models trained with these samples can only be used in similar situations, where the patient is trying to demonstrate their symptoms to another. Moreover, the questions posed in the interview must be carefully designed, and no noise interference is allowed. If we take into account factors such as group, language, and culture, the sample is often even less representative.

### 6.3. Small Sample Size

In terms of the number of subjects, all databases consist of a limited number of data samples because depression is a mental disorder and ethical issues are important for the publication of datasets. Due to the sensibility of depression speech and ethics problems, most institutions cannot obtain sufficient samples.

In addition, heterogeneity is a problem that makes cross-dataset validity difficult to establish. For example, aspects of imaging data such as data collection, scanning parameters, and processing methods hamper the generalization to other datasets. This makes it difficult to draw comparisons based on the results.

### 6.4. Simplified Limited Categories

In many datasets, the samples of mental disorders are divided into two categories: Depressive versus healthy people. This means that the randomization level of the classification is 50%. In the AVEC series, however, a continuous BDI score was used. However, this dataset focuses only on the dimension of depression, so essentially, it still selects only two categories of people (though with different severities of depression), a condition extremely inconsistent with real life. Outpatients come to the doctor with a wide variety of problems. Their symptoms are not standardized, and some symptoms of different illnesses can be relatively similar. When the diagnosis is strictly based on the DSM, it is extremely difficult to make such decisions. Moreover, there is the problem of co-morbidity. In cases of co-morbidity, clinicians also need to determine the dominant disorder. For example, depression with schizophrenia is a different diagnosis from schizophrenia with depression, and the medication will change.

Because AI learning uses artificial datasets with small sample sizes and lacks ecological validity, the “AI diagnosis” is actually a logical paradox and self-fulfilling prophecy. As a result, the accuracy of AI diagnoses is greatly reduced when faced with ecological clinical data.

## 7. Challenges from Annotations

### 7.1. Is the Annotation Valid?

In AVEC2013, the authors called for teams to participate in emotion and depression recognition using the data provided. The Depression Recognition Sub-Challenge requires participants to predict the level of self-reported depression as indicated by the BDI for every experiment session (i.e., one continuous value per multimedia file). This approach to annotation is difficult for psychiatrists to understand. The emotional status of a subject is normally hidden or cannot be displayed even without any display rules. It is nearly invalid to score the depression of a patient by a self-rating scale, simply by watching the performance of that person. In the BDI, many items are subjectively reported and not known to bystanders. In addition, depressive patients are not always in a depressed state. They may only have relatively obvious symptoms when they are having an episode, and many times they do not show symptoms that bystanders would recognize.

Data annotations serve as the basis for future work. However, the annotations may not be acceptable in many datasets. The models based on such annotations can therefore not be used in practice.

### 7.2. Can Subjective Feelings Be Mapped from Objective Measures?

In the diagnostic criteria for depressive disorders, subjective experiences or qualitative descriptions (i.e., high-level information) are often used. Predicting depression by measuring physiological signals or mannerisms (i.e., low-level information) implies that objectively measured signals can be used to map subjective feelings and qualitative descriptions; in other words, low-level information can be directly employed to map depression (i.e., top-level information) (see Figure 2 for the paths). For example, less smiling means that the individual is depressed, and a depressive disorder can be inferred from the depressed mood; thus, depression can be directly predicted by the amount of smiling. However, the amount of smiling is not necessarily related to emotion. The inconsistency of the connection between facial expressions and emotions has been much elaborated upon in previous research [49]. It would be inaccurate to argue that the amount of smiling can directly predict the presence of a depressive disorder. Similarly, physiological indicators such as heart rate, blood oxygen level, and skin conductance do not clearly point to high or low moods and thus are not directly related to depressive disorders. It is also not correct to use verbal and physical indicators to speculate about a person’s mood or whether they have a depressive disorder. Therefore, the fundamental challenge is that most objectively measured signals have no direct or even indirect relationship with depressive disorders.

## 8. Can Multimodality Be a Solution to the Complexity?

Many believe that since a single modality cannot be mapped to a subjective experience or qualitative description, supplementing multiple aspects of information through multimodality could be a more appropriate approach. From this view, substantial individual variances can be resolved, such as facial expressions being effective for A, voice for B, and heart rate variability for C. In such situations, multimodality could provide the key features required to predict depressive disorders. However, the signals must actually be valid and provide complementary rather than conflicting information. In reality, multimodal information may also provide noise and be conflicting. For example, some people in a depressive episode become irritable while others become depressed, often with quite different levels of expressiveness. If the prediction model is to adapt to different genders, ages, and personality traits and use different pathways for depression, the sample size required would be huge and the feasibility extremely low.

## 9. Conclusions

Due to the lack of objectivity in the diagnosis of mental illnesses, AI diagnoses are desired more than ever. This is because AI diagnoses are achieved from a large amount of objective data and are not influenced by personalized emotions, experiences, and other human traits. Although we can expect AI diagnosis to play a greater role in the future, there are still a number of difficulties with it today.

These difficulties arise from the data, such as capacity limitations of datasets and the ecological validity of training datasets. These technical difficulties can potentially be improved by purely technical improvements. However, another (and more fundamental) aspect of the difficulties of AI diagnosis comes from issues with diagnosing mental illness itself. Existing diagnostic criteria for mental disorders are still described in terms of relating symptoms to symptom labels. Most current symptoms are described subjectively and qualitatively, but AI training data are primarily based on low-level information (e.g., facial expressions, voice characteristics) and may predict high-level information (e.g., emotions) based on quantifiable low-level information or directly predict top-level information (e.g., depressive disorders). Despite the advantages of AI techniques in data collection and analysis, erroneous understanding of mental disorders may occur at the very beginning of AI system development.

Overall, the real challenge for AI-based mental disorder diagnosis is not a technical one, nor is it wholly about data, but rather our overall understanding of mental disorders in general.

## Figures and Tables

**Figure 1 diagnostics-13-00002-f001:**
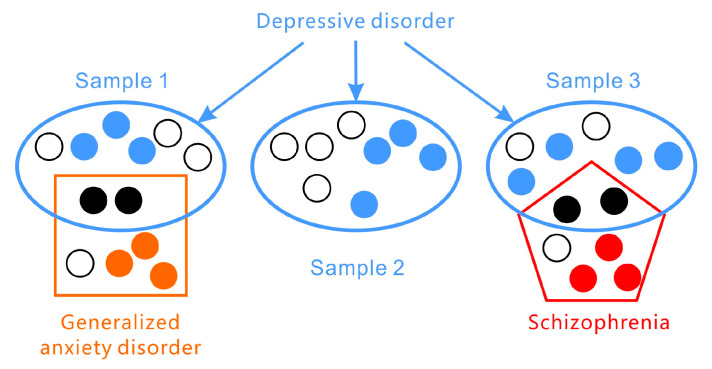
Depressive disorders vary greatly in their presentation of symptoms and are co-morbid with other disorders (such as generalized anxiety disorder and schizophrenia).

**Figure 2 diagnostics-13-00002-f002:**
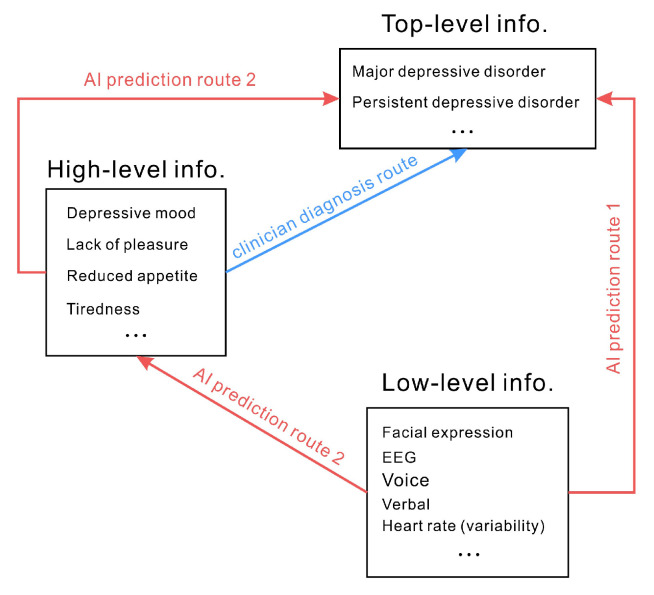
In the diagnostic criteria for depressive disorders, subjective experiences or qualitative descriptions (i.e., high-level information) are often used (clinician diagnosis route). AI generally tries to predict top-level with low-level information directly (route 1). There is also another possible route (route 2, not commonly used) that AI can predict high-level information first and then infer the top-level information.

**Table 1 diagnostics-13-00002-t001:** The four formats of conditional reasoning.

Logical Format	Diagnosis	Logical Value
DD *→DS ** Modus Pollens	If depressive disorder then depressive symptom	Valid
Non-DD→non-DS Denying the antecedents	If no depressive disorder then no depressive symptom	Invalid
DS→DD Affirming the Consequent	If depressive symptom then depressive disorder	Invalid
Non-DS→non-DD Modus Tollens	If no depressive symptom then no depressive disorder	Valid

* DD = depressive disorder; ** DS = depressive symptom.

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
