# Peer review of "Challenges for Artificial Intelligence in Recognizing Mental Disorders"

_diagnostics, 2022, doi:10.3390/diagnostics13010002_

Round 1
Reviewer 1 Report
Dear authors,
I would like to thank you for the opportunity to review this manuscript. Here are my comments:
Title: Replace “artificial intelligence” for AI
Abstract: Add “artificial intelligence” before AI
Line 25: to be added (AI) in parenthesis - like “Artificial intelligence (AI)”
Line 29-30: “Mental disorders …with the greatest focus on depression” (It’s incorrect to say that mental disorders focus on depression). Maybe you want to say that hundreds of articles focus on depression…to predict it
Line 35-36: to be added “machine learning methods” after EEG-based portable system
Line 42-44: One example from Ethiopia doesn’t reveal that the misdiagnosis rate is high
Line 45-48: The diagnosis of the mental disorders is established in the specialized psychiatrist setting. The misdiagnosis rates in the primary care settings are not relevant for argumentation.
Line 53-54: “…benefits…such as reducing manpower…”. I don’t think that could be a rational aim in medicine/psychiatry.
Line 55-57: “diagnosing mental disorders seems to be beneficial and likely soon replace psychiatrists”. Maybe it’s better to say “for helping psychiatrist to diagnose mental disorders”
Line 69-70: Not all the indicators found on depressive patients (line 65-69) are AI indicators: like biochemical indicators, neurotransmitters, EEG
Line 108: Maybe it’s better to name the subtitle “Challenges from analyzing depressive symptoms”
Line 155: Maybe it’s better to name the subtitle “Challenges from diagnosing depressive disorder”
Line 160-161: “Routine assessments include self-rating scales and clinician-based interviews”- Clinician-based interviews are doing in the first place. Maybe it’s better to discuss first about the clinician-based interviews, then to discuss about the self-rating scales.
Line 163: PHQ-9 is the abbreviation for…?
Line 197: AVEC2013 and AVEC2014 are the abbreviation for…?
Line 201: BDI-II is the abbreviation for…?
Line 294-298: The text from the Figure 2 must include “AI prediction route 2”
Line 299: It could disappear
Line 313-325: It could be moved after line 154 (Subtitle: Challenges from analyzing depressive symptoms)
Line 326-346: It could be moved after line 185 (Subtitle: “Challenges from diagnosing depressive disorder”).
Plus, you have to explain the abbreviation for “DD”, “DS”
Modus pollens or modus ponens?
Good luck!
Author Response
Thank you for your careful comments and suggestions. Please refer to the revised manuscript for the details.
- The title has been changed to “Challenges for Artificial Intelligence in Recognizing Mental Dis-orders”
- The abstract has started with “Artificial Intelligence (AI)”.
- I have re-defined mental disorders which is cited and correct the wording.
- As for the Ethiopia example, I cited directly from the paper: Ayano G, Demelash S, Haile K, et al. Misdiagnosis, detection rate, and associated factors of severe psychiatric disorders in specialized psychiatry centers in Ethiopia. Annals of general psychiatry 2021;20(1):1-10. The abstract displays this.
- According to the cited paper, they also used “the misdiagnosis rates in the primary care settings”. We cited this to demonstrate that there’s some mistakes in such settings.
- In many developing countries, there’s a lack of psychiatrists or psychiatric clinicians.
- We have replaced “such as reducing manpower” with “such as saving human resources”
- As suggested, we replaced “diagnosing mental disorders seems to be beneficial and likely soon replace psychiatrists” with “for helping psychiatrist to diagnose mental disorders”
- I’ve renamed the subheadings to make the whole structure look clearer.
- Many clinicians prefer to ask the patients to do the self-rating first and then do the clinical interview. Therefore, we explain self-rating first in the manuscript.
- We have already added the full names of the abbreviations. PHQ-9 = Patient Health Questionnaire-9; AVEC = The Continuous Audio/Visual Emotion and Depression Recognition Challenge; BDI-II = Beck Depression Inventory II; DD = depressive disorder; S = depressive symptom
- Figure 2 has included “route 2”.
- I have deleted session 6.2 “Large individual variances without consistent symptoms” and move the session “the logical fallacy of mental disorder diagnosis” to the place after “Challenges from standard diagnostic approaches”.
Thank you again for your supportive and constructive comments!
Reviewer 2 Report
1. The title of this article is the challenge of artificial intelligence in identifying mental disorders. Several questions should be included in the introduction:
1) There are problems in the identification of mental disorders.
2) The characteristics of artificial intelligence, the principle of how to identify mental disorders.
2. The structure of the paper needs to be revised.
In the text, the fusion recognition of auditory, visual, EEG signals, and multimodal features can be discussed separately, and the advantages and disadvantages of artificial intelligence in identifying emotional disorders can be summarized.
Author Response
Thanks for the reviewer's comments and suggestions. The followings are the replies:
- I have already provided a cited definition for mental disorder in the updated manuscript.
- I have tried to clarify Session 2 to make readers better understood in the updated manuscript, which is about "The rationale for using machine learning to identify mental disorders".
- I tried to give a clear description of the structure of the paragraphs in the updated manuscript.
Round 2
Reviewer 2 Report
This paper explores the challenges of artificial intelligence prediction in psychiatric disorders, with a comprehensive discussion that can be published.